# The Pivotal Role of TRP Channels in Homeostasis and Diseases throughout the Gastrointestinal Tract

**DOI:** 10.3390/ijms20215277

**Published:** 2019-10-24

**Authors:** Alessandro Alaimo, Josep Rubert

**Affiliations:** Department of Cellular, Computational and Integrative Biology (CIBIO), University of Trento, Via Sommarive 9, 38123 Povo (Tn), Italy

**Keywords:** transient receptor potential ion channels, gastrointestinal tract, sensory transduction, visceral hypersensitivity, inflammatory bowel disease, colorectal cancer, microbioma

## Abstract

The transient receptor potential (TRP) channels superfamily are a large group of proteins that play crucial roles in cellular processes. For example, these cation channels act as sensors in the detection and transduction of stimuli of temperature, small molecules, voltage, pH, and mechanical constrains. Over the past decades, different members of the TRP channels have been identified in the human gastrointestinal (GI) tract playing multiple modulatory roles. Noteworthy, TRPs support critical functions related to the taste perception, mechanosensation, and pain. They also participate in the modulation of motility and secretions of the human gut. Last but not least, altered expression or activity and mutations in the TRP genes are often related to a wide range of disorders of the gut epithelium, including inflammatory bowel disease, fibrosis, visceral hyperalgesia, irritable bowel syndrome, and colorectal cancer. TRP channels could therefore be promising drug targets for the treatment of GI malignancies. This review aims at providing a comprehensive picture of the most recent advances highlighting the expression and function of TRP channels in the GI tract, and secondly, the description of the potential roles of TRPs in relevant disorders is discussed reporting our standpoint on GI tract–TRP channels interactions.

## 1. Introduction

The primary function of the gastrointestinal (GI) tract is the transport, digestion, and absorption of nutrients and liquids. In addition, secretion of enzymes, regulation of homeostatic processes, appetite, bacterial catabolism, defense against toxins and pathogens, and formation of feces are daily carried out as well [1,2]. These regulations of GI functions depends on an appropriate set of several physiologic signals from the luminal contents to nervous, immune, and vascular systems [3,4]. However, the gut lumen does not stand alone, indeed the gut microbiota, composed of trillions of commensal bacteria, colonizes our intestine. These gut microbiota–diet interactions produce a large amount of bioactive metabolites that play crucial roles in the digestion of food, interaction with the gut epithelium, modulation of microorganisms, and development of the immune system [3,4]. The food we eat interacts with the gut epithelium, and, indirectly, with two general types of nerve fibers that broadly innervate the GI tract. The extrinsic afferent nerves originate from the dorsal, nodose and other ganglia, transport signals from the gut to the central nervous system. By contrast, the local nervous system, also known as enteric nervous system (ENS), establishes a widespread neural network enveloped in the wall of the gut. The ENS controls GI tract motility, peristalsis, ion and water secretion and detect chemical, osmolarity, or mechanical changes in the gut, among other functions [5]. As a result, the GI tract possess an elaborate and complex structure of molecular sensors that analyze the composition of food, modulate the secretory activity, and recognize adverse conditions in the lumen and wall of the alimentary canal.

Transient receptor potential (TRP) ion channel family are expressed all along the GI system where they regulate a plethora of functional features acting as molecular sensors and transducers [6,7]. In humans, the TRP family is composed of 28 members separated into six subfamilies (e.g., TRPC1-7, TRPV1-6, TRPM1-8, TRPA1, TRPML1-3, and TRPP1-2), which are mainly permeable to Na^+^ and Ca^2+^ and also allow moderate permeability to other physiological and toxic cations [8,9]. Recently, structural studies have revealed the architecture of TRP channels. These channels are homotetramers and each subunit consists of six transmembrane (TM) segments flanked by cytosolic N- and C-terminal domains with an ion permeable pore in the fifth and sixth TM segments [9,10,11,12]. They are extensively disseminated in the central and peripheral nervous systems, mainly in sensory neurons, but also in non-neuronal cells of mammalians. TRP channels act as molecular sensors of multiple and specific stimuli, for example, they can be gated by temperature, either heat (e.g., TRPV1) or cold (e.g., TRPA1 and TRPM8), taste, pH, osmolality, mechanical forces, bioactive substances (e.g., neurotransmitters and hormones), and by several natural and artificial ligands including pungent and cooling agents, fragrant components, dietary ingredients (e.g., chemicals and metabolites), environmental irritants, noxious substances, and inflammatory mediators [13,14,15]. Generally, these exogenous and/or local endogenous stimuli activate channels leading to plasma membrane depolarization and, eventually, to Ca^2+^ influx which prompts several signal transduction mechanisms (e.g., action potentials) which reach the autonomous and central nervous system, causing reflex responses and sensations [16,17]. Interestingly, a large number of disorders have been associated with TRP channel dysfunctions, such as anomalous expression levels, cellular relocation or mutations indicating their essential role in homeostasis and diseases [15,18,19,20].

This review describes the functions of TRP channel expressed in the GI tract, with special emphasis on their physiological roles. The implications of specific TRP channels on visceral taste, chemical and mechanical stimulation, as well as, their interconnections with gut microbial metabolites produced in the alimentary canal is also discussed. Lastly, we summarize what is known on the involvement of TRPs in GI diseases/disorders and the potential use of pharmacological regulators in the treatment of these malignancies.

## 2. Expression and Pathophysiological Roles of TRP Channels in the GI Tract

In the vast majority of cases, the TRP family are expressed by primary afferent sensory neurons that arise from ganglia and ENS neurons which, conjunctly, innervate the GI tract [21,22]. However, recent studies have also demonstrated the presence of TRP channels in GI non-neuronal cells, as enterocytes and enteroendocrine cells (including enterochromaffin cells) [7,23]. One of the most studied members of the TRP family is TRPV1, which is well-known as the capsaicin receptor, the irritating compound of hot chili peppers [10,24]. TRPV1 is largely expressed in the GI tract, mainly in primary afferent sensory neurons and in ENS neurons, but also in mucosal epithelial cells and enteroendocrine cells [25,26]. TRPV4, another member of the vanilloid subfamily, has been described to be expressed by primary afferent sensory neurons, epithelial cells, and infiltrated inflammatory cells in the intestine [27,28]. TRPA1 is one of the most promiscuous chemosensors, as it can be activated by a large number of dietary and noxious molecules such as isothiocyanates (horseradish, mustard oil, wasabi, etc.) and allicin (garlic), prostaglandins, exogenous irritants and products of oxidative stress [29]. This channel is expressed by ganglia and enteric primary neurons which project to the gut, and by mucosal cells [21]. Finally, two members of the TRPM subfamily are broadly presents in the GI tract. TRPM5 channels are located on the gut chemosensory cells, and lingual taste buds receptors cells [30,31]. On the other hand, TRPM8 is mostly expressed by primary afferent neurons which operates as “cold chemosensor”, being activated by cold temperatures and several chemicals such as menthol, geraniol, and linalool, to name only a few [32,33,34,35] (Figure 1).

From a cellular point of view, TRP channels display, at least, three functional roles in the GI tract: (i) molecular sensors and primary transductors of chemical and physical stimuli, (ii) effectors of other ion channels and receptors, and (iii) conventional ion channel, enabling the transport of cations across the plasma membrane [36,37]. For example, TRPA1, TRPM5/8, and TRPV1/4 recognize and transduce the signals coming from metabolites and noxious toxins contained in food, playing a key role in the intrinsic control of GI functions, in particular chemesthesis, taste, mechanosensation, visceral thermo- and nociception, and pain [38,39]. In addition, TRP channels are involved in secretory mechanisms of the digestive tract, modulate the alimentary canal motility, contribute in the absorption of Ca^2+^ and Mg^2+^ and, finally, are implicated in the control of membrane potential and excitability of neurons and epithelial cells [40,41].

Mutations or changes in the expression or function of TRPs can provoke diseases of the GI system, indicating that the achievement of a threshold of TRP function is critical to preserve normal physiological activity. These dysfunctions include, among others, pain and heartburn, visceral hypersensitivity, irritable bowel syndrome, inflammatory bowel diseases, gastroesophageal reflux disease, intestinal fibrosis, hypomagnesemia, hypocalcemia, and GI cancers. In the next sections, we summarize the current findings regarding the role of TRP proteins in the pathogenesis of some of the most prevalent GI disorders in human (Table 1).

### 2.1. Visceral Hypersensitivity and Irritable Bowel Syndrome

Sensory signals are dependent on the activities of various ion channels expressed by sensory neurons. The input signals generating in the gut are transported by primary sensory neurons, with their cell body in the dorsal root ganglion, and then, as action potentials, to second-order neurons of the spinal cord. This sensorial information is eventually transmitted to neuronal centers in the thalamus and the brain leading to conscious perception [86,87].

Irritable bowel syndrome (IBS) is the most recurrently diagnosed disorder by gastroenterologists worldwide, affecting more than 10% of the Western population [88,89,90]. IBS is a group of gastrointestinal symptoms characterized by chronic or recurrent abdominal pain and/or discomfort associated with altered defecation pattern without any evidence of an organic damage [90,91]. Frequently, people affected by IBD also present other disorders, such as chronic fatigue syndrome, anxiety, and major depression. The main mediator of the increased perception of pain arises from the gut and experienced by IBS patients is the Visceral Hypersensitivity (VH), which is defined as enhanced perception of noxious stimulus (hyperalgesia) and altered sensation of normal physiological stimuli to the abdominal region (allodynia) [92,93]. The pathogenesis of VH is complex and multifactorial involving neuronal, immune, and endocrine signaling pathways, but also can result from aberrant signaling from the gut to the central nervous system or vice versa [89]. Although the exact pathophysiological mechanism of VH is not fully understood, the upregulation and/or sensitization of nociceptors, including opioid and cannabinoid receptors and, in particular, TRP channels, is recognized to play a pivotal role in altered pain signaling in patients affected by IBS [6,56,94,95,96].

So far, TRPV1, TRPV4, TRPA1, and TRPM8 channels have been investigated for their implication in visceral pain. Evidence suggests that TRPV1 may be involved in pain and hyperalgesia [42]. In this frame, biopsies from patients affected by IBS reported that nerve fibers expressing TRPV1 in colon was notably increased in number and this increase correlated with pain severity [43,44]. Moreover, TRPV1 expression is improved in preclinical models of VH [45], whereas sensitivity to colorectal pain is decreased by TRPV1 antagonists and reduced in *Trpv1* knock-out mice [46,47]. It has been also demonstrated that the upregulation and sensitization of TRPV1 causing hyperalgesia is influenced, at least in part, by inflammatory mediators and endocrine factors produced by the gut microbiota such as histamine, 5-hydroxytryptamine (5-HT) and nerve growth factor (NGF) [45,48,49]. For instance, NGF can raise the levels of TRPV1 and promote its insertion into the plasma membrane of peripheral nerve endings [49]. Strikingly similar to TRPV1, manifold evidences support that TRPV4 activation triggers VH, playing a significant role in hyperalgesia [42,57]. Expressed in the extrinsic primary afferents, TRPV4 potentiate the pro-algesic effects of histamine and 5-HT [56] and co-localize with protease-activated receptor-2 (PAR-2) which mutually cooperate causing mechanical hyperalgesia in the mouse colon [58,59]. The excitatory effect of TRPV4 on colonic afferents is potentiated by PAR-2 agonists [60], while the nociceptor sensitivity is alleviated by application of the TRPV4 antagonist HC067047, further emphasizing the potential role of TRPV4 in VH [61].

Recent studies have recognized that TRPA1 participates in inflammatory responses and the establishment of mechanical and chemical colonic hypersensitivity [21,75]. TRPA1 stimulation induces the Ca^2+^-mediate secretion of substance P (SP) and calcitonin gene-related peptide (CGRP) and other transmitters from afferent nerve fiber, which eventually results in inflammation and VH [76]. Several murine models of IBS present increased TRPA1 function in sensory ganglia and colon [77]. The same applies to humans, where the upregulation of TRPA1 in biopsies of IBD patients has been observed [78]. Meseguer et al. reported that bacterial lipopolysaccharides (LPS) also may activate TRPA1 channels inducing acute gut inflammation and visceral pain associating microbiome – gut epithelium interactions with the potential role of microorganisms and microbial metabolites [79]. These observations suggest that TRPA1 may be involved in the progression and subsistence of VH [80,81].

Contrary to TRPV1, TRPV4, and TRPA1, TRPM8 activation may attenuate pain through suppression of TRPV1 activity and the inhibition of TRPA1 function [68,69]. The analgesic effects mediated by the activation of TRPM8 can alleviate neuropathic and VH perception or mitigating cold hypersensitivity in inflammatory and nerve-damage [42,70]. Numerous studies have demonstrated that TRP channels are also co-expressed on sensory neurons and frequently have been found upregulated or sensitized in biopsies from IBS patients, suggesting an interconnection among TRPs. As a matter of fact, almost all TRPA1-positive sensory neurons co-express TRPV1 and the gating of TRPA1 can modulate TRPV1 activity [97]. Moreover, intraperitoneal co-injection of a TRPV1 inhibitor and a TRPA1 antagonist in a mouse model of experimental colitis results in a substantial reduction of VH compared to injection of the antagonists separately, suggesting a synergistic effect [97]. These data suggest that the interconnection of TRP channels can assist to pain sensitivity in the GI tract and may serve as novel therapeutic targets, which may potentially be associated with the diet–gut microbiota interactions.

### 2.2. Inflammatory Bowel Diseases

Trauma or diseases occurring along the GI tract may cause tissue damage leading to the release of inflammatory mediators that activate local neurons and other cells. The secretion of inflammatory molecules is mainly triggered by fibroblasts, epithelial and immune cells surrounding the injured tissue. Thus, sensory neurons provoke inflammatory response, sensing sites of inflammation and promoting protective or dolorous behaviors.

Inflammatory bowel diseases (IBDs) are the most common chronic and relapsing disorders of the GI tract, characterized by severe inflammation resulting from a complex interaction between genetic and environmental factors [98,99]. The major clinical symptoms of IBD comprise abdominal pain, bloody diarrhea, gastrointestinal bleeding, weight loss, fever, and anemia. IBD includes two main subtypes: Crohn’s disease (CD) and ulcerative colitis (UC) [100]. CD affects the whole intestine, as well as the mouth, esophagus, stomach, and the anus, while UC affects only the mucosa of the colon and the rectum. IBDs can be a painful and debilitating diseases with potentially life-threatening complications and poor quality of life for patients [101]. This explains the growing interest in understanding the complex pathophysiological mechanisms of IBD, the identification of the key mediators implicated in the disorders and, eventually, the finding of new therapeutic targets [102]. The involvement of TRP channels has been widely studied in the context of gut inflammation [28,44,76] (Figure 2).

Pro-inflammatory molecules, such as prostaglandins, bradykinin, or ATP, can sensitize TRPV1 and increase the probability of channel gating by heat and agonists (i.e., capsaicin) [103]. Colonic biopsies from patients affected by functional inflammatory disorders (CD and UC) present upregulation of TRPV1 in nociceptive afferent neurons. In this way, it was demonstrated that in the mouse colon TRPV1 enables an upregulated release of the neuropeptides CGRP and SP [44,50]. Because of the fact that TRPV1 mainly operates as an intermediary transducer, interceding in visceral nociception and intestinal inflammation after gut epithelial injury [51,52]. Mounting evidence indicates that TRPV4 exerts a pro-inflammatory role in the GI tract. TRPV4 was found overexpressed in inflamed gut tissue of human CD and, mostly, UC patients and its activation induces somatic and visceral pain [57,62,63]. Further in vitro studies on epithelial expressed TRPV4 have also demonstrated that the channel activation triggers IL-8 production in mucosa cells, suggesting that TRPV4 may play a role in gut inflammation via a non-neuronal mechanism [64]. Analogously to TRPV1, the activation of TRPA1 on visceral sensory neurons induces the secretion of neuropeptides, with consequent vasodilatation, inflammation and hyperalgesia in the gut [81,82]. Likewise, TRPA1 is unequivocally overexpressed in models of colitis in mice and in colon biopsies obtained from human UC and CD patients [78]. However, different lines of evidence have disclosed a protective role of TRPA1 against gut inflammation. TRPA1 may facilitate the restoration of tissue damages in IBD. This helpful effect seems due to the activation of sensory TRPA1 channels but also could involve those expressed in gut epithelial and immune cells, which antagonize the pro-inflammatory activities of sensory TRPV1 channels via neuropeptide release [83]. Although little is known about the TRPM8 involvement in IBD, it has been proposed that the activation of TRPM8 elicit an anti-inflammatory effect via inhibition of neuropeptides release [104].

### 2.3. Intestinal Fibrosis

Fibrosis is an essential reparative mechanism that occurs after tissue injury or damage. In physiological conditions, fibrosis is an extremely coordinated process, however, when the inflammation persists, myofibroblasts continuously secrete extracellular matrix factors affecting the architecture of the tissue. Therefore, fibrosis is a frequent outgrowth of chronic inflammation related to terminal diseases of many vital organs such as lung, liver, heart, and intestines, being a major mortality risk in developed countries [105].

Intestinal fibrosis is the major complication of IBD [106,107]. The massive stratification of fibrotic tissue increases the consistency of the bowel wall, thus, restraining the elasticity and the function of the affected area. Current anti-inflammatory therapies have had a limited efficacy against fibrosis in CD patients [108]. For this reason, it is highly required to find new treatments taking into account mechanisms of fibrosis [107,109,110].

An intricate set of cytokines, growth factors, and chemokines are involved in the recruitment, differentiation, activation, and regulation of myofibroblasts, therefore, with the intestinal fibrogenesis [111,112]. The main player of this polypeptides’ cocktail is TGF-β, which supports multiple aspects of myofibroblast functions in fibrogenesis such as transformation, proliferation, migration, resistance to apoptosis, stress fiber formation, and collagen synthesis [107,113]. TGF-β and its receptors are up-regulated in the inflamed intestines of IBS patients [114,115,116] and, interestingly, several TGF-β pathways have relevant connections to intracellular Ca^2+^ homeostasis and dynamics which, in turn, are mediated by some members of the TRP superfamily [107,117,118]. For example, TRPC3, another member of the TRP superfamily, plays a crucial role in the progression of fibroproliferative disorders including renal [119] and myocardial fibrosis [120,121]. On the other hand, it has been described that TRPA1 [84] and TRPC6 [72] channels are involved in human intestinal fibrogenesis. Recently, results indicate that the activation of TRPA1 expressed in myofibroblast, in addition to its anti-inflammatory function in IBD [83], might have significant therapeutic relevance in preventing fibrosis [85]. On the other hand, TRPC6 channels have gained growing importance in the development of fibrosis, especially as mediators of myofibroblast function [73]. It was found that TGF-β stimulation increased TRPC6 expression in fibroblasts and that the inhibition of calcineurin, a downstream mediator of TRPC6-dependent Ca^2+^ signaling, resulted in reduced myofibroblast differentiation [72]. Similar results have been obtained in vitro treating InMyoFibs human intestinal myofibroblast cells with TGF-β. These cells respond to TGF-β stimulation increasing the production of α-smooth muscle actin (α-SMA) stress fiber along with improved synthesis and functionality of TRPC6 channels [74]. These evidences are consistent with the assumption that increased TRPC6 channel activity is needed for TGF-β-mediated myofibroblast differentiation and fibrosis development. However, in contrast with the above results, the in vitro knockdown of TRPC6 or its pharmacological inhibition also increases the levels of anti-fibrotic IL-10 and IL-11 and the expression of collagen in myofibroblast [74]. These discordant effects of TGF-β related to myofibroblast TRPC6 channel are difficult to elucidate with a single mechanism, suggesting that TRPC6 may be involved in fibrosis in a very elaborate way. Although more exhaustive investigation is necessary to decipher the interconnection between TGF-β and TRPC6, these results contributed to our knowledge about the pathogenesis of intestinal fibrosis and may be beneficial to establish a new therapeutic strategy for chronic IBDs such as CD [71].

## 3. TRP Channels in Colorectal Cancers

Colorectal cancer (CRC), the most prevalent cancer of the GI tract, represents the third most commonly diagnosed malignancy worldwide in men and women and the second largest cause of death in Europe related to cancer [122,123]. Growth factors, oncogenes and receptors play a critical role in the tumorigenesis, and the propagation of CRC. However, other environmental and genetic factors play significant roles in the development of CRC [124]. Over the last two decades, accumulating evidence has demonstrated that several members of the TRP family channels show altered expression and activity in cancer cells [125,126]. Levels of expression of TRPC, TRPM, and TRPV proteins depend on the cancer stage. Those differences have been recognized mainly in melanoma, glioma and prostate, breast, kidney, and bladder cancers, suggesting an oncogenic role for some TRPs while others may function as tumor suppressors [18,127,128]. TRPs activity implicates many aspects of cancer development or “hallmarks of cancer”, such as tumor migration and invasion, exaggerated cell proliferation, cell survival, angiogenesis, and enhanced resistance to cell death [129,130,131].

To date, TRPV1 and TRPV6 have been exclusively recognized as relevant in CRC carcinogenesis. As mentioned above, the TRPV1 channel is also involved in IBD, a chronic disorder strongly correlated with risk of CRC [53]. Animal models have demonstrated a protective role for TRPV1 in the tumorigenesis of colon cancer, indeed TRPV1- Knockout mice exhibited lower expression of anti-inflammatory neuropeptides and greater occurrence and number of CRC compared with controls [54]. In accordance with these data, Sung and colleagues demonstrated that agonist-mediated activation of TRPV1 induced apoptosis of CRC cells [55]. On the other hand, an overexpression of TRPV6 in SW480 colorectal cancer cell lines and colon carcinomas have been associated with an enhanced proliferation of malignant cells [67]. In a recent report, Pérez-Riesgo et al. have assessed differential gene expression of all human TRP channels by Next Generation Sequencing (NGS) to investigate differences in expression of several genes involved in intracellular Ca^2+^ transport in CRC [65]. The results obtained, comparing human normal colonic cells versus human colon cancer lines, revealed that only a dozen of the TRPs are expressed in colonic cells, either normal or tumor. The data also indicated that TRPV6 was the unique TRP gene significantly overexpressed in CRC cells, resembling the previously reported enhanced levels of TRPV6 in prostate cancer [66]. Notwithstanding, the functional consequences of changes in expression of TRPV1 and TRPV6 channels in CRC remain to be established.

## 4. Experimental Models and Therapeutic Opportunities

The implication of TRP channels in pathophysiological mechanisms within the GI tract, and the changes in TRPs expression and activity related with VH, IBD, and CRC have attracted enormous interest in the therapeutic exploitation of these proteins [14,132,133]. Growing evidence points out TRP channels as potential targets for novel analgesics effective in GI pathophysiological conditions including VH, IBS, and IBD, thus, over the past years several corporations started research screening to identify TRP modulators. Multiple approaches can be used to counteract dysfunctional TRP channels, such as blocking TRPs with antagonists, desensitizing the channel with agonists or using drugs that prevent channel trafficking and incorporation in the plasma membrane [134].

As discussed earlier, TRPV1 activity is crucial in colorectal mechanosensation and neurogenic inflammation, representing a captivating target for pharmacological treatment of IBS. Several clinical trials have been implemented with selective TRP channel antagonists, including some very promising drugs that have been patented [134,135]. For instance, the TRPV1 antagonist JYL1421 has proved to be effective to counteract VH caused by various stimuli, colitis and the increase in TRPV1 immunoreactivity in DRG [136]. There have been a large number of studies using animal models, mainly mice, to investigate the implication of TRPV1 in IBDs and VH. For example, the 2,4,6-trinitrobenzenesulfonic acid (TNBS)–induced colitis mice model was used to demonstrate that the upregulation of TRPV1 in ganglia neurons triggered chemical and mechanical VH [136]. Moreover, mice lacking TRPV1 failed to develop post-inflammatory VH following acute colitis induced by dextran sulfate sodium (DSS) [137]. Finally, in rat stress-induced models, visceral pain was decreased administrating TRPV1 antagonists [138].

Theoretically, TRPA1 could be an excellent target for the treatment of GI inflammatory disorders and VH, however, studies on TRPA1 antagonists and their efficacy and safety are very low [139,140]. In mice, intracolonic administration of TRPA1 agonists enhanced the visceromotor response [81]. In TNBS-induced colitis mice models, the overexpression of TRPA1 in colonic DRGs led to an increased visceromotor response to colorectal distention. This effect may be prevented treating mice with a TRPA1 small interfering RNA (siRNA) [141] or TRPA1 antagonists [97]. Moreover, in DSS colitis models pretreated with a TRPA1 antagonist, the TRPA1 agonist Allyl isothiocyanate (AITC) induced VH [52]. Interestingly, the combination of a TRPV1 antagonist (BCTC) with a TRPA1 antagonist (TCS-5861528) in a rat model of colitis resulted in a substantial reduction of VH compared to injection of the drugs separately, suggesting a synergistic effect [97].

TRPM8 plays a primary role in the treatment of VH and GI pain as demonstrated by the analgesic effect of its agonists. Menthol, a well-known TRPM8 activator, was used during decades in traditional medicine to alleviate abdominal spasms and visceral pain [142]. In addition, modern clinical trials showed that the ingestion of peppermint-oil in IBS patients significantly decreases abdominal pain perception [143,144] and is effective in the treatment of functional dyspepsia [145]. Preclinical models suggest that the activation of TRPM8 results in a reduced GI pain perception. Transgenic mice lacking TRPM8 showed loss of innocuous cold sensation, diminished responses to noxious cold temperatures, and a reduced response to cooling compounds. For instance, the administration of a mixture of TRPM8 agonists peppermint and caraway oil reduced the colonic hypersensitivity in TNBS-induced models [146]. It has been reported that in clinical trials, the administration of enteric-coated peppermint oil decreased visceral pain and increase the life quality of IBS patients [143,144]. Moreover, TRPM8 activation by its agonist icilin alleviate inflammation in experimental colitis models and reduced the levels of inflammatory cytokines and mucosal injury, probably due to an inhibition of neuropeptide release [104].

The potential use of two TRPV4 antagonists in the treatment of VH and colitis have been suggested in recent studies. First, the TRPV4 antagonist HC067047 has had results beneficial to attenuate the mechanosensitivity perception of VH in ex vivo experiments [61]. Also, the results obtained with the antagonist RN173 injected systemically and administered locally was suggestively encouraging, suggesting that the blockage of TRPV4 may be helpful in attenuation of IBD [62]. Promising results have also been obtained employing live imaging of rectal biopsies [147]. Balemans and colleagues found that the administration of the agonist GSK1016790A triggered an increased Ca^2+^ response in submucosal neurons of IBS patients probably due to TRPV4 desensitization [147]. The critical role of TRPV4 in VH was analyzed using various preclinical models. For example, the agonist 4α-phorbol 12,13-didecanoate (4α-PDD) induced significant TRPV4-mediated Ca^2+^ influx in isolated colonic DRG neurons [57]. On the other hand, VH responses of colonic afferent fibers were increased by the agonist 5,6-EET (an endogenous arachidonic acid metabolite) and considerably reduced by TRPV4 antagonists [58]. Finally, in vivo models demonstrated that the administration of supernatants derived from IBS biopsies induced VH in mice, while, conversely, silencing TRPV4 in mouse primary afferent neurons inhibited the hypersensitivity induced by IBS biopsies supernatants [56].

Altogether, these experimental data indicate the crucial role of TRP channels in mechanosensitivity, chemosensitivity, and pain in GI pathological states. However, it should be considered that targeting TRP channels is a challenging therapeutic approach, given their widespread expression and homeostatic roles outside the GI system [51,148]. Accordingly, even though there are some positive reports, the direct inhibition or activation of TRP channels often leads to severe off-target adverse effects. Unfortunately, the direct drugs targeting of TRP channels present some limitations due to their abundant distribution in the body, although other drawbacks must be considered. For example, the activation of TRPV1 channels have a noticeable hyperthermic effect that limits the clinical usefulness of their agonists. Analogously, TRPA1 and TRPM8 activation often are associated to hypothermic side-effects. Moreover, several TRP agonists and inhibitors are not highly specific and cross-reactivity between different TRP channels have been observed. Finally, the expression of specific GI variant and short isoforms further difficult the targeting of these channels.

In recent years, a very promising alternative to the direct inhibition of TRP channels has emerged. Mounting evidence have proved that ω-3 polyunsaturated fatty acid–derived resolvins (Rv) are potent anti-inflammatory agents that preclude the activation of TRPA1, TRPV1, and TRPV4 channels [149,150]. Resolvins are endogenous lipids produced mainly by eosinophils and neutrophils, effective as anti-inflammatory molecules [151]. So far, RvE1, RvD1, and RvD2 are the most studied for their analgesic properties and mounting evidence indicates that Rv strongly interferes indirectly with TRP channels activity [151]. Thus, Rv selectively inhibited acute pain evoked by administration of TRPV1, TRPV4, and TRPA1 agonists and attenuate inflammatory hypersensitivity to various stimuli [152,153]. Although the mechanism by which Rv inhibit TRP channels is not entirely known, these molecules represent a very interesting and promising alternative to resolve VH and GI inflammatory pain [153].

## 5. Conclusions

Herein, we briefly reviewed these progresses underscoring the important role of TRPs as cellular sensors and transductors of chemical, thermal, and toxic stimuli within the alimentary canal. Our comprehension of the involvement of TRP channels to gastrointestinal taste, sensation, and nociception has considerably increased in the recent past. Some gastrointestinal disorders, including VH, IBS, IBD, and CRC, are characterized by an increased expression of TRP channels and/or change of its function, proof of a direct implication of TRPs in these pathologies. Several studies have demonstrated that these channels are potentially druggable targets in GI dysfunctions and new promising molecules have been discovered. However, future basic research should be focused on deciphering the relationship between the recently disclosed structure-function data and the signaling pathways related to TRP channels for the development of more selective drugs for the treatment of GI pathologies. Alternatively, a comprehensive understanding of gut microbial metabolites in homeostasis and in diseases may also attenuate GI disorders.

## Figures and Tables

**Figure 1 ijms-20-05277-f001:**
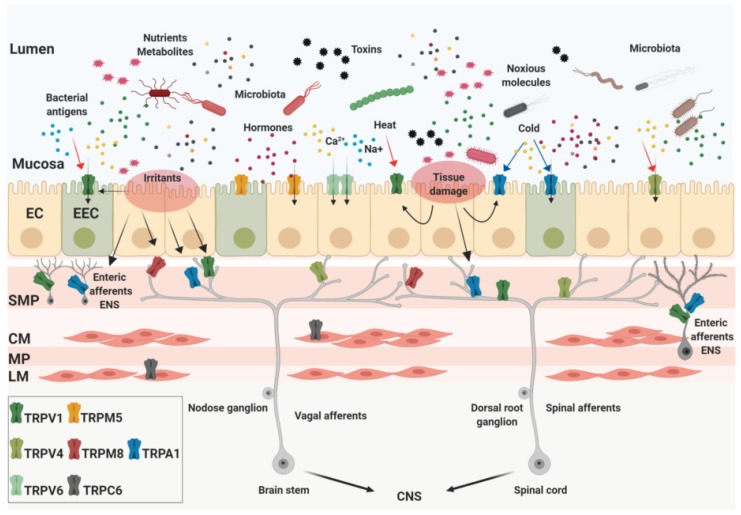
Expression of the transient receptor potential (TRP) channels in the gut. The intestinal lumen is densely populated by nutrients, gut microbial metabolites, chemicals, and toxins that can be present in the food ingested, released by the intestinal tissue or produced by bacteria. The gastrointestinal (GI) tract mucosa is mainly composed by epithelial cells (enterocytes, EC) and enteroendocrine cells (for instance enterochromaffin cells, EEC) whose apical membranes express several TRP channels. They act as sensors of specific stimuli such as temperature, pH, osmolality, mechanical stress, and recognize bioactive molecules. In addition to the mucosa, TRPs are located in almost all layers of the intestinal wall. Some of them are expressed by afferent neurons in the myenteric (MP) and submucosal plexus (SMP) as part of the intrinsic enteric nervous system (ENS). Two layers of muscle tissue, the circular (CM) and the longitudinal (LM) also express TRP channels. Finally, TRP channels expressed by extrinsic afferent nerves are implicated in the critical mechanism of visceral perception, transporting signals from the gut to the central nervous system (CNS). Created with BioRender.com.

**Figure 2 ijms-20-05277-f002:**
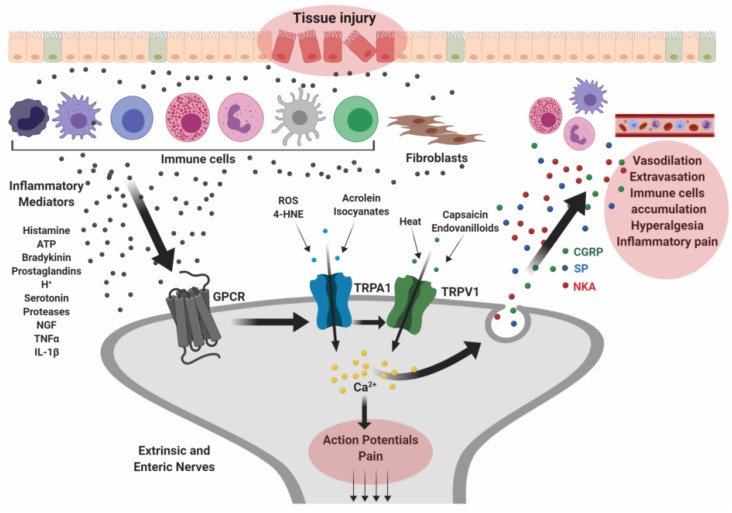
Implications of TRP in the pathophysiology of inflammatory bowel diseases. Tissue damage occurring along the GI mucosa may cause release of inflammatory mediators that activate principally immune cells and fibroblasts surrounding the injured tissue. These cells release more inflammatory molecules, including ATP, histamine, tumor necrosis factor α (TNF-α), interleukin-1β (IL-1β), nerve growth factor (NGF), and proteases, which bind specific G protein-coupled receptors (GPCRs) expressed in primary afferent sensory neurons. The consequent activation of TRP channels trigger the release of neuropeptides such as substance P (SP), calcitonin gene-related peptide (CGRP), and neurokinin A (NKA) which promote neutrophil accumulation, vasodilation, pain and other inflammatory responses. Created with BioRender.com.

**Table 1 ijms-20-05277-t001:** TRP channels in the alimentary canal.

	Ligands	Physical Stimuli	Endogenous Stimuli	GI Tract Expression	GI Related Disorders
**TRPV1**	Capsaicin (red pepper) Piperine (black pepper) Gingerol (ginger) Resiniferatoxin (*Euphorbia poissonii*)	T ≥ 43 °C Voltage Distension pH	Cannabinoids Anandamide Eicosanoids Acid Bradykinin Serotonin Histamine Proteases	Sensory neurons Enteric neurons Epithelial cells Enteroendocrine cells [25,26]	IBS VH [42,43,44,45,46,47,48,49] IBD [44,50,51,52] CRC [53,54,55]
**TRPV4**	4-α Phorbol (*C. tiglium*) Bisandrographolide A (*Andrographis paniculata*)	T ≥ 25 °C Mechanical Osmolarity Distension pH	Anandamide Eicosanoids Bradykinin Citrate Arachidonic acids Histamine Proteases	Sensory neurons Epithelial cells [27,28]	IBS VH [42,56,57,58,59,60,61] IBD [57,62,63,64]
**TRPV6**		Voltage		Epithelial cells [65,66]	CRC [65,66,67]
**TRPM5**	Steviol glycosides (*Stevia rebaudiana*) Rutamarin (*Ruta graveolens*)	Voltage pH	Intracellular Calcium Arachidonic acid Phospholipase C Sugars Acid	Epithelial cells Enteroendocrine cells Taste receptor cells [30,31]	
**TRPM8**	Menthol (mint) Linalool (laurels, cinnamon, rosewood) Geraniol (geranium, rose oil) Eucalyptol (*Eucalyptus*)	T ≤ 25 °C Voltage	Cannabinoids Anandamide Lysophospholipids Polyunsaturated fatty acids Bradykinin Prostaglandins Serotonin	Sensory neurons [32,33,34,35]	IBS VH [42,68,69,70]
**TRPC6**	Hyperforin and Adhyperforin (*Hypericum perforatum*)	Mechanical	Arachidonic acid Lysophospholipids Eicosanoids	Smooth muscle cells [71]	IF [71,72,73,74]
**TRPA1**	Allicin (garlic) Carvacrol (oregano) Cinnamaldehyde (cinnamon) Diallyl disulfide (garlic) Gingerol (ginger) Ally isothiocyanate (mustard horseradish, wasabi)	T ≤ 18 °C Mechanical Distension pH	Cannabinoids Bradykinin Nicotine Prostaglandins Histamine Proteases ROS 4-HNE	Sensory neurons Enteric neurons Epithelial cells Enteroendocrine cells [21]	IBS VH [21,75,76,77,78,79,80,81] IBD [78,81,82,83] IF [84,85]

Abbreviations: IBS—Irritable Bowel Syndrome; VH—Visceral Hypersensitivity; IBD—Inflammatory Bowel Diseases (Crohn’s disease and ulcerative colitis); IF—Intestinal Fibrosis; CR—Colorectal Cancer; ROS—Reactive oxygen species; 4-HNE—4-Hydroxynonenal.

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
