# Peer review of "The Pivotal Role of TRP Channels in Homeostasis and Diseases throughout the Gastrointestinal Tract"

_ijms, 2019, doi:10.3390/ijms20215277_

Round 1
Reviewer 1 Report
Overall, this is a concisely written review on TRP channels in the digestive tract. The authors cover a broad number of channels in multiple pathological conditions and this makes information covered in the review somewhat diffuse. It is suggested that that there be more focused discussion either on select channels or disease states.
There are minor grammatical and spelling corrections listed below.
line 49 - "...ion channels" should be "...ion channel"
line 50 - "where regulate" should be "where they regulate"
line 56 - "ions" should be "ion"
line 69 - "channels" should be "channel"
line 96 - "...neurons where" should be "...neurons which"
line 96 - "activate" should be "activated"
line 121 - "Remarkably" is not required to start the sentence.
line 128 - Table 1 requires specific reference numbers against TRP expression data and related disease conditions.
line 166 - "release" is not required.
line 171 - "...where mutually" should be "...which mutually"
line 182 - "...gut acute" should be "...acute gut"
line 194 - "VHS" should be "VH"
line 200 - "...activate locally" should be "...activate local"
line 205 - "condition" is not required.
line 238 - "trigger" should be "triggers"
line 313 - "cols" should be "colleagues"
Author Response
Reviewer 1
Overall, this is a concisely written review on TRP channels in the digestive tract. The authors cover a broad number of channels in multiple pathological conditions and this makes information covered in the review somewhat diffuse. It is suggested that that there be more focused discussion either on select channels or disease states.
We thank reviewer 1 for his/her time spend on evaluation of our manuscript and the helpful suggestions made. The entire manuscript has been checked for proper grammar and changes were made accordingly (in blue).
There are minor grammatical and spelling corrections listed below.
We corrected these typos and grammatical mistakes. Thank you very much. The reviewer can find the corrections (in green) in the revised text
line 49 - "...ion channels" should be "...ion channel" Amended (line 48)
line 50 - "where regulate" should be "where they regulate" Amended (line 49)
line 56 - "ions" should be "ion" Amended (line 55)
line 69 - "channels" should be "channel" Amended (line 67)
line 96 - "...neurons where" should be "...neurons which" Amended (line 92)
line 96 - "activate" should be "activated" Amended (line 92)
line 121 - "Remarkably" is not required to start the sentence. Deleted (line 117)
line 128 - Table 1 requires specific reference numbers against TRP expression data and related disease conditions.
As required, Table 1 has now the specific reference numbers. Thank you for the suggestion (page 4)
line 166 - "release" is not required. Deleted (line 161)
line 171 - "...where mutually" should be "...which mutually" Amended (line 165)
line 182 - "...gut acute" should be "...acute gut" Amended (line 177)
line 194 - "VHS" should be "VH" Amended (line 190)
line 200 - "...activate locally" should be "...activate local" Amended (line 196)
line 205 - "condition" is not required. Deleted (line 201)
line 238 - "trigger" should be "triggers" Amended (line 233)
line 313 - "cols" should be "colleagues" Amended (line 310)
Reviewer 2 Report
This is an interesting review on a topical subject that, in my opinion, is publishable, pending some (mostly minor) modifications
Major Comments
1) I believe this review would benefit if a separate section or a paragraph in section 4 is devoted on some of the challenges interpreting trp channel studies in vivo. For example many of the antibodies used are not as specific as claimed and the same is true for some agonists and inhibitors. Additionally, targeting one trp isoform could lead to compensatory upregulation of a related trp variant. I believe these issues should be highlighted.
2) Some stylistic/spelling mistakes throughout. I have highlighted some that I have noticed in the minor comments section. Please correct as appropriate.
3) In the section about fibrosis (2.3) please also include references implicating TRPC3 in cardiac and renal fibrosis (e.g. Seo et al. (2014) PNAS 111 (4) 1551-1556 and Saliba et al. (2015) JASN 26 (8) 1855-1876).
4) Some of the general references are rather dated. For example, the general references about fibrosis and TGF (106 and 110) are from 2010-2011. Please, where applicable, include more recent references.
5) Please, where possible include the original basic science article instead of a review article (for example reference 122).
Minor Comments
1) Please correct Ca2+ to Ca2+ throughout (e.g. lines 52 and 65) same for Na+.
2) The sentence in lines 68-72 is not very clear please re-phrase.
3) Line 143 "occidental population". Generaly, in terms of style, I would prefer less complicated use of language - e.g replace this by western population.
4) Line 177 "Ca2+ - mediated"
5) Lines 261-263: This sentence is not very clear. Please re-phrase.
Author Response
Reviewer 2
This is an interesting review on a topical subject that, in my opinion, is publishable, pending some (mostly minor) modifications
We thank reviewer 2 for his/her time spend on evaluation of our manuscript and the helpful suggestions made.
The reviewer can find the corrections/changes required in the revised text (in red)
Major Comments
1) I believe this review would benefit if a separate section or a paragraph in section 4 is devoted on some of the challenges interpreting trp channel studies in vivo. For example many of the antibodies used are not as specific as claimed and the same is true for some agonists and inhibitors. Additionally, targeting one trp isoform could lead to compensatory upregulation of a related trp variant. I believe these issues should be highlighted.
We thank the referee for this constructive suggestion. As required, section 4 has been changed following the reviewer indications. We have referred to literatures and papers, re-analyzed the data and reconstructed this section to improve the quality of our paper. Obviously, this was beyond the scope of this review, however we think that the section 4 is more complete now, and we thank again the referee for bringing this to our attention. We try our best to revise it and we hope these efforts will be appreciated.
2) Some stylistic/spelling mistakes throughout. I have highlighted some that I have noticed in the minor comments section. Please correct as appropriate.
Amended, thank you. We have also reviewed the manuscript and fix as many typos that we were able to recognize (in blue), and corrected typos and grammatical mistakes as suggested by referee 1 (in green)
3) In the section about fibrosis (2.3) please also include references implicating TRPC3 in cardiac and renal fibrosis (e.g. Seo et al. (2014) PNAS 111 (4) 1551-1556 and Saliba et al. (2015) JASN 26 (8) 1855-1876).
Amended, thank you. As suggested, the references have been added in lines 266-268: “For example, TRPC3, another member of the TRP superfamily, plays a crucial role in the progression of fibroproliferative disorders including renal [107] and myocardial fibrosis [108,109]”
4) Some of the general references are rather dated. For example, the general references about fibrosis and TGF (106 and 110) are from 2010-2011. Please, where applicable, include more recent references.
References 106 and 110 have been eliminated. More recent references are now integrated in the text. For example, references 95 and 101-104 (lines 263-264)
5) Please, where possible include the original basic science article instead of a review article (for example reference 122).
This have also been amended. The old Ref 122, has been replaced for Ref 111: Davis et al. A TRPC6-Dependent Pathway for Myofibroblast Transdifferentiation and Wound Healing In Vivo. Dev. Cell 2012, 23, 705–715. We also included more original articles, replacing some reviews. Thank you very much
Minor Comments
1) Please correct Ca2+ to Ca2+ throughout (e.g. lines 52 and 65) same for Na+.
Amended. Thank you
2) The sentence in lines 68-72 is not very clear please re-phrase.
As required the sentence has been re-phrased (lines 66-68). Thanks
3) Line 143 "occidental population". Generaly, in terms of style, I would prefer less complicated use of language - e.g replace this by western population.
We have removed “occidental” and replaced it with “western”. Thank you for the suggestion
4) Line 177 "Ca2+ - mediated"
Amended. Thank you
5) Lines 261-263: This sentence is not very clear. Please re-phrase.
We have rephrased the sentence (lines 256-258). Thank you
Round 2
Reviewer 1 Report
The revisions are acceptable as is the manuscript for publication.